# Connectionist Temporal Classification with Maximum Entropy Regularization

**Hu Liu    Sheng Jin    Changshui Zhang**
Institute for Artificial Intelligence, Tsinghua University (THUAI)
Beijing National Research Center for Information Science and Technology (BNRist)
State Key Lab of Intelligent Technologies and Systems
Department of Automation, Tsinghua University, Beijing, P.R.China
{liuhu15, js17}@mails.tsinghua.edu.cn   zcs@mail.tsinghua.edu.cn

## Abstract

Connectionist Temporal Classification (CTC) is an objective function for end-to-end sequence learning, which adopts dynamic programming algorithms to directly learn the mapping between sequences. CTC has shown promising results in many sequence learning applications including speech recognition and scene text recognition. However, CTC tends to produce highly peaky and overconfident distributions, which is a symptom of overfitting. To remedy this, we propose a regularization method based on maximum conditional entropy which penalizes peaky distributions and encourages exploration. We also introduce an entropy-based pruning method to dramatically reduce the number of CTC feasible paths by ruling out unreasonable alignments. Experiments on scene text recognition show that our proposed methods consistently improve over the CTC baseline without the need to adjust training settings. Code has been made publicly available at: https://github.com/liuhu-bigeye/enctc.crnn.

## 1   Introduction

Past few years have witnessed significant progress in sequence learning tasks. Currently recurrent neural network (RNN) with Connectionist Temporal Classification (CTC) [5] has become a popular framework and is widely used in areas such as speech recognition [6, 8, 1, 2, 22], sign language recognition [4], video segmentation [10, 18] and scene text recognition [29, 19, 7]. CTC views the outputs of RNN as a probability distribution over all possible alignments and directly learns the mapping from input sequences to target sequences. It is proven to be effective in weakly supervised sequence modeling with only temporal order supervision but no alignment information provided.

CTC can be regarded as a kind of Multiple Instance Learning (MIL) [21]. From the perspective of MIL, the label sequence is a bag containing all feasible paths. CTC learns by maximum likelihood estimation (MLE) over the summation of all feasible path probabilities. However, as the number of feasible paths grows exponentially with the input sequence length, it is hard for CTC to find the most suitable one. More seriously, once CTC finds a dominant feasible path during the training process, the error signal will concentrate on the vicinity of this path, and the prediction of this feasible path will continuously strengthen until this path completely dominates the prediction output. As blanks are included in most of the feasible paths, dominant paths are often overwhelmed by blanks, interspersed by sharp spikes (narrow regions along the time axis) of non-blank labels, which is known as the CTC peaky distribution problem [5, 22].

This problem of CTC will lead to the following consequences.

- Harm the training process. The positive feedback-like error signal means CTC lacks exploration and is prone to fall into worse local minima.

- Output overconfident paths. CTC tends to concentrate all its output distribution over one specific path. On the one hand, it is not suitable to handle the situation where the segmentation boundary is ambiguous, e.g. adjacent syllables in speech recognition and action switching in continuous sign language recognition. On the other hand, the low-entropy output distribution is a symptom of overfitting [31], leading to low prediction accuracy.

- Output paths with peaky distribution. The peaky distribution is not desirable for sequence segmentation tasks when the model needs to densely predict labels for each time-step. Even if only the temporal order of labels is required, learning the correct segmentation will improve model generalization and interpretation abilities.

Motivated by the maximum entropy principle [13], we propose a maximum conditional entropy based regularization for CTC (EnCTC). We consider the conditional distribution of feasible paths given the input sequence and label sequence. EnCTC prevents the entropy of feasible paths from decreasing too fast, thus alleviating the impact of CTC's positive feedback-like error signal and encouraging exploration during training. It also prevents the probability from being dominated by a single path and solves the peaky distribution problem. It mitigates overfitting and is more suitable for depicting ambiguous segmentation boundaries.

We further consider another solution to the problem of finding the reasonable feasible path—limit the size of the feasible set. We observe that in many sequence learning tasks, the intervals between the adjacent elements in the input sequence are almost the same, *e.g.* characters in text recognition, syllables in speech recognition and gestures in continuous sign language recognition. We summarize this phenomenon as the equal spacing prior. Therefore, we propose an algorithm to limit the size of the CTC feasible set by eliminating these unreasonable paths that seriously violate the equal spacing prior (EsCTC). Moreover, the equal spacing prior can give theoretical explanations from the perspective of maximum entropy, which indicates that equal spacing is the best prior without any additional subjective assumptions.

The main contributions of this paper can be summarized as: (1) We propose a maximum conditional entropy regularization for CTC (EnCTC), which encourages exploration for CTC training and prevents peaky output distributions. (2) We derive from equal spacing prior a pruning algorithm (EsCTC) to effectively limit the size of CTC feasible set and give theoretical explanations from the perspective of maximum entropy. (3) We provide polynomial-time dynamic programming algorithms for calculating EnCTC, EsCTC and their combination (EnEsCTC). (4) We validate the proposed methods on scene text recognition tasks and show that these methods are able to improve the baseline model without changing training settings.

## 2 Related Work

CTC [5] is a popular framework for end-to-end sequence learning tasks, such as speech recognition [6, 8, 1, 2, 22], scene text recognition [29, 19, 7], sign language recognition [4] and video segmentation [10, 18]. However, CTC tends to output highly peaky distribution [5, 22], which is a sign for model overconfidence. To remedy this, some intuitive smoothing methods are proposed. [32] introduces path sampling to speed up CTC training, which has a side-effect of reducing the posterior spikiness. [18, 22] smooth the estimation of path label priors by discounting the majority number of "background" labels and increasing the counts of rare actions, then normalizing the posterior distribution using the priors when decoding. However, their smoothing methods are performed during decoding and do not affect the training process.

Model overconfidence is a universal problem in the fields of machine learning and several regularization approaches have been proposed to handle it. Weight decay [16] regularize by limiting the range of parameters; Dropout [9] and DropConnect [33] by adding noise to the model structure; Stochastic Pooling [37] by adding noises to the pooling operations. Their methods regularize model parameters, but do not directly regularize the output distributions.

The maximum entropy based regularization [23] has long been investigated in the literature to regularize model behavior. The maximum entropy estimation [13] involves no additional assumptions when estimating the distribution. It assigns the positive weight to every possible situation and produces

the maximum entropy predictions under certain constraints. In reinforcement learning, [36, 25], the maximum entropy regularization is proposed to encourage exploration and prevents early convergence. In supervised learning, [27] proposes to penalize the entropy of high-confidence output softmax distributions by adding negative entropy to the objective function.

This work is mostly related to those CTC based methods that use regularization to reduce the over-confident prediction. In [3], label smoothing [31] and increasing temperature of the SoftMax function are employed to improve beam search. [15] simply adds confidence penalty regularization term [27] to regularize the output distribution. To some extent, their approaches improve generalization and reduce the peaky distribution. However label smoothing [31] and confidence penalty [27] regularize the model prediction at each time-step, which corresponds to regularize both feasible and invalid paths. We instead regularize the entropy among the feasible paths of CTC to handle peaky distribution problem. Alignment constraints on CTC paths have been previously explored in [28, 32]. We further propose an entropy based pruning method to reduce the searching space of possible alignments and facilitate convergence.

## 3 Method

### 3.1 Problem Definition

Throughout the paper, the sequence learning problem is defined as follows:

- The dataset consists of pairs of input sequences $X$ and corresponding target sequences $l$.
- Each element of the target sequence is defined in a fixed-length label alphabet $L$.
- Each input sequence $X$ should be longer than its corresponding target sequence $l$.
- The alignment between $X$ and $l$ is unknown but in a sequential manner.

### 3.2 Connectionist Temporal Classification (CTC)

CTC [5] is a popular method for sequence learning. It enables the end-to-end model training with no pre-defined alignment information required. In the framework of CTC, given an input sequence $X_{1:T}$ of length $T$, the model predicts a sequence $y^{1:T}$ of length $T$, where $y^t$ denotes the probability vector of observing labels over the fixed-length label alphabet $L'$. $L' = L \cup \emptyset$ contains all the pre-defined labels including a 'blank' label $\emptyset$ at time-step $t$. We call the concatenation of observed labels at all time-steps as a path $\pi$.

In order to access the relationship between path $\pi$ and target sequence $l$, CTC defines a many-to-one mapping operation $\mathcal{B}$. $\mathcal{B}$ firstly removes the repeated labels then removes all blanks from the given path. Given a label sequence $l$, we define feasible paths as all those $\pi$ that can be mapped onto $l$ through $\mathcal{B}$. The conditional probability of a given target sequence is defined as the sum of probabilities of all feasible paths.

$$p(l|X_{1:T}) = \sum_{\pi \in \mathcal{B}^{-1}(l)} p(\pi|X_{1:T}), \tag{1}$$

where the probability of $\pi$ is defined as

$$p(\pi|X_{1:T}) = \prod_{t=1}^{T} y_{\pi_t}^t, \forall \pi \in L'^T. \tag{2}$$

CTC guides the end-to-end model training by directly optimizing the loss function $L_{ctc} = -\log p(l|X_{1:T})$. It uses dynamic programming to efficiently sum up all the feasible paths.

However, due to the massive amount of feasible path in Equation 1, directly optimizing CTC loss may not result in a good alignment. More specifically, the error signal of CTC loss with respect to $y_k^t$ is computed as:

$$\frac{\partial L_{ctc}}{\partial y_k^t} = -\frac{1}{p(l|X) y_k^t} \sum_{\{\pi | \pi \in \mathcal{B}^{-1}(l), \pi_t = k\}} p(\pi|X). \tag{3}$$

We can see that the error signal is proportional to the fraction of all feasible paths that go through symbol $k$ at time $t$. That means once a feasible path is dominant, the error signal of $y^t_{\pi_t}$ will dominate $y^t$ at all time-steps $t$, causing all the probabilities to focus on a single path while ignoring its alternatives. This positive feedback-like error signal makes CTC lack exploration during training and tend to overfit.

### 3.3 Maximum Conditional Entropy Regularization for CTC (EnCTC)

In order to facilitate training and be more likely to find the feasible paths, we propose a regularization method based on maximum conditional entropy (EnCTC). EnCTC constitutes an entropy-based regularization term that prevents the entropy of the feasible paths from decreasing too fast, leading to better generalization and exploration.

$$L_{enctc} = L_{ctc} - \beta H(p(\pi|l, X)), \tag{4}$$

where $\beta$ controls the strength of the maximum conditional entropy regularization.

The entropy of the feasible paths given input sequence $X$ and target sequence $l$ is defined as:

$$
\begin{aligned}
H(p(\pi|l, X)) &= - \sum_{\pi \in \mathcal{B}^{-1}(l)} p(\pi|X, l) \log p(\pi|X, l) \\
&= -\frac{1}{p(l|X)} \sum_{\pi \in \mathcal{B}^{-1}(l)} p(\pi|X) \log p(\pi|X) + \log p(l|X).
\end{aligned}
\tag{5}
$$

Fast convergence to one feasible path means that the entropy term of Equation 5 reduces rapidly.

More specifically, the error signal of entropy regularization term $-H(p(\pi|l, X))$ with respect to $y^t_k$ is computed as:

$$\frac{\partial -H(p(\pi|l, X))}{\partial y^t_k} = \frac{Q(l)}{p(l|X)y^t_k} \left( \frac{\sum_{\{\pi|\pi \in \mathcal{B}^{-1}(l), \pi_t = k\}} p(\pi|X) \log p(\pi|X)}{Q(l)} - \frac{\sum_{\{\pi|\pi \in \mathcal{B}^{-1}(l), \pi_t = k\}} p(\pi|X)}{p(l|X)} \right), \tag{6}$$

where $Q(l) = \sum_{\pi \in \mathcal{B}^{-1}(l)} p(\pi|X) \log p(\pi|X)$. We can see from Equation 6 that this error signal is proportional to the fraction of $p(\pi|X) \log p(\pi|X)$ minus the fraction of $p(\pi|X)$ for all feasible paths that go through symbol $k$ at time $t$. Notice that these two fractions are zero when $p(\pi|X) = 0$, but $p(\pi|X) \log p(\pi|X)$ decreases more rapidly around zero and reaches its minimum when $p(\pi|X) = 1/e$. Therefore, paths with probability between 0 and $1/e$, *i.e.* paths near the dominant path, contribute the most to the error signal. This error signal will in turn increase the probability of the nearby paths and improve the exploration during training.

### 3.4 Equal Spacing CTC (EsCTC)

As discussed previously, CTC is prone to output degenerated paths due to large searching space. In this section, we present a pruning method to dramatically reduce the searching space of possible alignments.

We observe that in many sequence learning tasks, such as scene text recognition, the spacing of two consecutive elements (or the width of elements) is nearly the same. We assume that this property will hold for all reasonable alignments. We thus propose to measure the spacing equality and set a threshold to rule out those unreasonable alignments. This protects our model from distractions and helps convergence.

Here we provide some theoretical demonstration of the equal spacing prior based on maximum entropy over segmentation.

Given an input sequence $X_{1:T}$ of length $T$ and a label sequence $l$, define the segmentation sequence $z_{1:|l|}$ splitting $X$ into a series of short segments. Each segment corresponds to a duration $z_s$ and label $l_s$, $\sum_{s=1}^{|l|} z_s \leq T$ (with an optional all-blank suffix). Sequence $Z$ consists of the starting positions of each segment, where $Z_s = \sum_{i<s} z_i$. Feasible paths that map onto $l$ through $\mathcal{B}$ can be split into feasible paths satisfying different segmentation sequence $z$, with $\pi_{Z_{s+1}-1} = l_s$ and

$\mathcal{B}(\pi_{Z_s:Z_{s+1}}) = l_s$ for each segment. We denote feasible paths that satisfy the segmentation sequence $z$ as $\mathcal{B}_z^{-1}(l)$. Therefore, we have $\dot{\bigcup}_z \mathcal{B}_z^{-1}(l) = \mathcal{B}^{-1}(l)$, where $\dot{\bigcup}$ is the union operation on pairwise disjoint subsets.

The entropy of feasible paths satisfying a specific segmentation sequence is defined as

$$H(p(\pi|z,l,X)) = -\sum_{\pi \in \mathcal{B}_z^{-1}(l)} p(\pi|z,l,X) \log p(\pi|z,l,X). \tag{7}$$

**Theorem 3.1.** *Among all segmentation sequences, the equal spacing one has the maximum entropy.*

$$\underset{z}{\operatorname{argmax}} \max_p H(p(\pi|z,l,X)) = z_{es}. \tag{8}$$

*Proof.* As to the definition of $\mathcal{B}_z^{-1}$, the feasible path of each segment must start with several blanks, followed by the element corresponding to that segment more than once. Therefore, we can find the maximum of entropy for every segment

$$\max_p H(p(\pi_{Z_s:Z_{s+1}-1}|z,l,X)) = \log z_s. \tag{9}$$

According to Equation 2 and the definition of segmentation, the entropy of feasible paths satisfying a specific segmentation sequence can be disassembled into the summation of the entropy of segments.

$$\begin{aligned}
\max_p H(p(\pi|z,l,X)) &= \sum_{s=1}^{|l|} \max_p H(p(\pi_{Z_s:Z_{s+1}-1}|z,l,X)) \\
&= \log \prod_{s=1}^{|l|} z_s.
\end{aligned} \tag{10}$$

With the constraint $\sum_{s=1}^{|l|} z_s \le T$, it can be derived from Jensen inequality that equidistant segmentation maximized the entropy term. $\qquad\square$

In the case of strict equal spacing, the length of each element in the segmentation sequence is equal and sums to $T$. As the real-world sequence learning tasks do not strictly satisfy the equal-spacing constraints, we introduce a slight relaxation by requiring each segment being no longer than $\tau$ ($1 \le \tau \le |l|$) times the average segmentation length, including the remaining suffix.

$$C_{\tau,T} = \{z | T - \tau\frac{T}{|l|} \le \sum_{s=1}^{|l|} z_s \le T, z_s \le \tau\frac{T}{|l|}\}. \tag{11}$$

Thus the Equal Spacing CTC deformation (EsCTC) guides the end-to-end model training by optimizing the loss function:

$$\begin{aligned}
L_{esctc} &= -\log p_\tau(l|X) \\
&= -\log \sum_{z \in C_{\tau,T}} \sum_{\pi \in \mathcal{B}_z^{-1}(l)} p(\pi|X).
\end{aligned} \tag{12}$$

We consider the EsCTC as complementary to EnCTC that explicitly rules out all unreasonable alignments. Therefore, we also combined the Equal Spacing CTC deformation with maximum conditional entropy regularization defined in section 3.3.

$$L_{enesctc} = L_{esctc} - \beta H(p_\tau(\pi|l,X)). \tag{13}$$

### 3.5 Algorithm and Complexity Analysis

EnCTC, EsCTC and EnEsCTC can be efficiently calculated by dynamic programming. Due to space limit, the details of the dynamic programming are presented in the supplementary material.

Parallelizing all independent computing units, the time complexity of CTC and EnCTC forward-backward dynamic programming is $O(T)$. The time complexity of EsCTC and EnEsCTC is $O(\frac{T^2\tau}{|l|})$, which is between $O(\frac{T^2}{|l|})$ and $O(T^2)$ depending on $\tau$. The space complexity of CTC and EnCTC is $O(T|l|)$ since forward and backward variable are kept for gradient computing. The space complexity of EsCTC and EnEsCTC is $O(T^2|l|)$.

## 4   Experiments

We evaluate our proposed method on several standard benchmarks for scene text recognition tasks. We use CRNN [29] as our baseline model. CRNN trains a fully-convolutional network (FCN) with 2 bidirectional LSTM layers on the top. The model is optimized by CTC loss. For all the experiments, we use the same training settings and parameters as CRNN [29], except for the proposed regularization term.

### 4.1   Datasets and Evaluation Metrics

When compared with the state-of-the-art approaches, we follow the standard experimental settings [29, 19, 17, 30] to train on the synthetic dataset (Synth90K) [11] once and test on four challenging real-world benchmarks for scene text recognition without fine-tuning. In such experimental settings, the model is purely trained with synthetic text data, but tested on real-world scene text datasets, which requires generalization across datasets. The four real-world benchmarks include ICDAR-2003 (IC03) [20], ICDAR-2013 (IC13) [14], IIIT5k-word (IIIT5k) [24] and Street View Text (SVT) [34] datasets. **Synth90K [11]** consists of 8M training images and 1M testing images generated by a synthetic data engine. **ICDAR-2003 (IC03) [20]** test set consists of 251 full scene images and 860 cropped image patches containing words. We follow the standard evaluation protocol as [34, 11, 35] to only consider words with alphanumeric characters and at least three characters. **ICDAR-2013 (IC13) [14]** extends IC03 and contains 1015 groundtruth cropped word images from real scenes. **IIIT5k-word (IIIT5k) [24]** is a real-world scene text recognition dataset with 3,000 cropped word images downloaded from Google Image Search. **Street View Text (SVT) [34]** test dataset contains 249 Google Street View images, from which 647 word images are cropped for testing. We further evaluate the model generalization performance on a small dataset **Synth5K**. Synth5K is a small-scale dataset with 5K training data and 5K testing data randomly sampled from Synth90K.

For all the experiments, sequence accuracy is used as the evaluation metric, *i.e.* the percentage of testing images correctly recognized. We only evaluate the model in lexicon-free mode, that the predictions are made only based on the input image without any predefined lexicon.

### 4.2   Implementation Details

We use RMSProp to train our model and set the batch size to 100. The learning rate is fixed at $1 \times 10^{-3}$ during training. The training stops at 150 epochs. Models and loss functions are all implemented using Pytorch [26]. Computations of the forward and backward variables are carried out in log-space to avoid overflow.

For EnCTC, we varied the strength term $\beta$ in the range [0.1, 0.2, 0.5] and found 0.2 work best. For EsCTC, we varied the equal-spacing term $\tau$ in the range [1.1, 1.3, 1.5, 1.7, 2.0, 2.5] and found 1.5 work best. We simply apply the same parameters to the EnEsCTC algorithm. For all the experiments, we set $\beta$ as 0.2 and $\tau$ as 1.5 without further tuning.

### 4.3   Qualitative Analysis

In this section, we provide some qualitative analysis. All the experiments are performed on Synth5K.

**Error Signal in Training**

Figure 1 visualizes the evolution of the model prediction and error signal. In the early stage of training (on the top), the prediction is near uniform distribution. Since the entropy term has zero gradient for uniform distribution, the error signal of EnCTC coincides exactly with CTC. We also see that EsCTC and EnEsCTC produce more concentrated error signal, that is, the algorithms consider that

the element at the head of the label sequence cannot appear at the rear of the input sequence, reducing the size of feasible path set. In the interim training period (shown on the bottom), we observe that the error signal of EnCTC gets smoother than that of CTC, which helps to enhance exploration during training and avoid the peaky distribution. EsCTC tends to pull the blue and green labels closer to get more equally spaced, producing more reasonable alignment. Error signal of EnEsCTC has the properties of both algorithms.

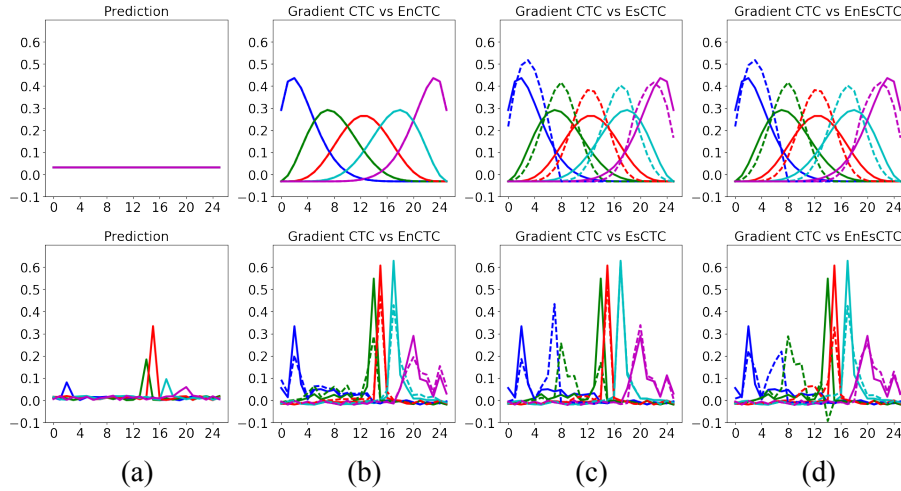

(a)           (b)           (c)           (d)

Figure 1: The evolution of prediction and error signal. Colors indicate different labels and the horizontal axis correspond to the position in the input image. (a) the prediction of CTC model (b)(c)(d) the gradient of EnCTC, EsCTC and EntEsCTC versus CTC respectively, in which the solid lines correspond to CTC and the dashed lines to the three proposed algorithms. **Top:** Initial training stage. **Bottom:** Interim training period. The CTC predicts a correct label sequence in this case.

**Alignment Evaluation**

We observe from Figure 2 that CTC outputs a highly peaky distribution and shows inferior alignment to the three proposed algorithms. It is attributed to that CTC lacks exploration during training and considers equally for the massive amount of feasible paths. EnCTC strengthened the exploration during CTC training process, producing a better and more reasonable alignment where each label occupies a certain range along the horizontal axis. The probabilities overlap at the junction of two consecutive characters, effectively modeling the boundary ambiguity. EsCTC eliminates unreasonable CTC feasible paths based on the equal spacing prior, which results in a more accurate alignment. The alignment result of EnEsCTC shows it combines the advantage of both EnCTC and EsCTC.

**Pruning Analysis for EsCTC**

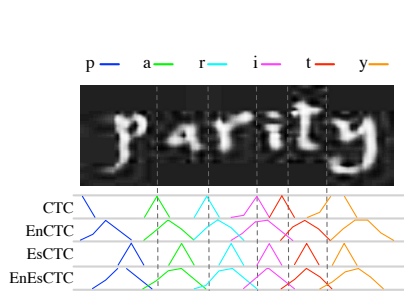

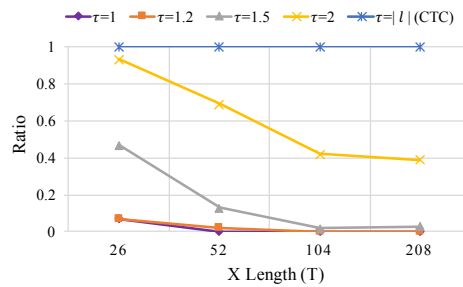

Figure 2: Illustration of the prediction results of an example image. The vertical dotted line indicates the ground truth segmentation results.

Figure 3: The influence of $\tau$ on path pruning for different data scales.

Figure 3 illustrates the influence of $\tau$ (see Equation 11) on path pruning for different data scales. The x-axis is the length of the input sequence, while the y-axis is the ratio of feasible path numbers of EsCTC to that of CTC. The length of the label $|l|$ is 12 in the experiment. We see that $\tau$ ($1 \leq \tau \leq |l|$) controls the searching space reduction rate. Note that when $\tau = 1$, EsCTC requires strict equality of spacing; when $\tau = |l|$, EsCTC degenerates to original CTC. As $\tau$ approaches 1, the ratio of remaining paths quickly drops to zero, indicating the path pruning effect of EsCTC. We also observe that as the length of input sequence ($T$) increases, the path pruning effect gets more significant. Please refer to the supplementary material for more details.

## 4.4  Evaluation of Model Generalization Performance

We further evaluate the model generalization performance on Synth5K. Note that over 97% of the words in the training set only appear once and 94% of words in the test set does not appear in the training set. Therefore, the model needs strong generalization ability. We also compare with two regularization methods, namely label smoothing (LS) [31] and confidence penalty (CP) [27].

As shown in Table 1, our proposed regularization methods show superior performance over LS and CP. LS and CP directly regularize the model prediction at each time-step and improve the generalization of the CTC baseline. However, their methods indiscriminately regularizes both feasible and unfeasible paths. EnCTC instead regularizes the entropy of feasible paths, which is more reasonable for CTC. When combined with EsCTC, our final EnEsCTC achieves the best generalization performance.

## 4.5  Comparisons with the State-of-the-art Methods

We compare our proposed methods with several state-of-the-art approaches. STAR-Net [19] and CRNN [29] are CTC-based methods, while R2AM [17] and RARE [30] are attention-based methods. STAR-Net [19] employs a much deeper network than [29] to extract features. STAR-Net [19] and RARE [30] also apply STN [12] to remove the distortions. Note that we use the same training settings and parameters as our baseline CRNN [29], except only for the proposed regularization term. Experimental settings and implementation details are summarized in Section 4.1 and Section 4.2.

Table 2 presents the results of the scene text recognition task. We find that our method outperforms the baseline CRNN without changing the training settings. It also achieves comparable performance to the state-of-the-art methods with a much simpler architecture. In particular, our proposed methods achieve the best performance on IC03 and IC13 datasets.

Table 1: Evaluation of model generalization.

| Method | Synth5K |
| --- | --- |
| CTC | 38.1 |
| CTC + LS [31] | 42.9 |
| CTC + CP [27] | 44.4 |
| EnCTC | 45.5 |
| EsCTC | 46.3 |
| EnEsCTC | **47.2** |

Table 2: Comparisons with the state-of-the-art methods.

| Method | IC03 | IC13 | IIIT5K | SVT |
| --- | --- | --- | --- | --- |
| CRNN [29] | 89.4 | 86.7 | 78.2 | 80.8 |
| STAR-Net [19] | 89.9 | 89.1 | **83.3** | **83.6** |
| R2AM [17] | 88.7 | 90.0 | 78.4 | 80.7 |
| RARE [30] | 90.1 | 88.6 | 81.9 | 81.9 |
| EnCTC | 90.8 | 90.0 | 82.6 | 81.5 |
| EsCTC | **92.6** | 87.4 | 81.7 | 81.5 |
| EnEsCTC | 92.0 | **90.6** | 82.0 | 80.6 |

## 5  Conclusions

In this paper, we have presented a novel maximum entropy based regularization for CTC (EnCTC), which maintains reasonable possibilities among all the feasible paths, to enhance generalization and exploration. Moreover, we derive from equal spacing prior a pruning algorithm (EsCTC) to effectively reduce the space of the feasible set and give theoretical explanations from the perspective of maximum entropy. The experiments on scene text recognition benchmarks demonstrate that our proposed methods achieve superior performance than the baseline and show better generalization ability. Our proposed method achieves comparable performance to the state-of-the-art methods with a much simpler architecture. We believe that the proposed regularization is general, which can be used to consistently improve the performance of the original CTC model.

**Acknowledgments**

This work is supported by NSFC (Grant No. 61876095, No. 61751308 and No. 61473167) and Beijing Natural Science Foundation (Grant No. L172037).

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
