[Supplementary Material]

# Supplementary Material for Connectionist Temporal Classification with Maximum Entropy Regularization

**Hu Liu    Sheng Jin    Changshui Zhang**
Institute for Artificial Intelligence, Tsinghua University (THUAI)
Beijing National Research Center for Information Science and Technology (BNRist)
State Key Lab of Intelligent Technologies and Systems
Department of Automation, Tsinghua University, Beijing, P.R.China
{liuhu15, js17}@mails.tsinghua.edu.cn  zcs@mail.tsinghua.edu.cn

We present the dynamic programming algorithms for calculating EnCTC, EsCTC and EnEsCTC.

Following the marks defined in CTC [1], given an input sequence $X_{1:T}$ of length $T$, the model predicts a sequence $y^{1:T}$ of length $T$, where $y^t$ denotes the probability vector of observing labels over the fixed-length label alphabet $L'$ at timestep $t$. $L' = L \cup \emptyset$ contains all the pre-defined labels including a 'blank' label $\emptyset$ .

Observed labels at all timesteps are concatenated in a path $\pi$, where $p(\pi) = \prod_{t=1}^{T} y_{\pi_t}^t$. Define a many-to-one mapping operation $\mathcal{B}$ that firstly removes the repeated labels then removes all blanks from the given path. All feasible paths satisfying $l$ are defined as $\{\pi | \pi \in \mathcal{B}^{-1}(l)\}$. We also have $p(l|X) = \sum_{\pi \in \mathcal{B}^{-1}(l)} p(\pi|X)$ which is CTC's optimization goal.

## 1 EnCTC

The entropy term in EnCTC can be converted to

$$
\begin{aligned}
H(p(\pi|l, X)) &= - \sum_{\pi \in \mathcal{B}^{-1}(l)} p(\pi|X, l) \log p(\pi|X, l) \\
&= - \sum_{\pi \in \mathcal{B}^{-1}(l)} \frac{p(\pi|X)}{p(l|X)} \log \frac{p(\pi|X)}{p(l|X)} \\
&= -\frac{1}{p(l|X)} \sum_{\pi \in \mathcal{B}^{-1}(l)} p(\pi|X) \log p(\pi|X) + \log p(l|X).
\end{aligned}
\tag{1}
$$

Note that $p(l|X)$ can be calculated by CTC forward-backward algorithm. We only need to provide dynamic programming algorithms for calculating $Q(l) = \sum_{\pi \in \mathcal{B}^{-1}(l)} p(\pi|X) \log p(\pi|X)$.

### 1.1 The EnCTC Forward-Backward Algorithm

Consider a modified label sequence $l'$ that adds blank before, after and in between each labels in $l$, we get $|l'| = 2|l| + 1$.

Define forward variable $\gamma(t, s)$ as the total $p \log p$ satisfying $l'$ prefix of length $s$ till time t.

$$
\begin{aligned}
\gamma(t, s) &\triangleq \sum_{\{\pi_{1:t} | \mathcal{B}(\pi_{1:t}) = \mathcal{B}(l'_{1:s}), \pi_t = l'_s\}} p(\pi_{1:t}|X) \log p(\pi_{1:t}|X) \\
&= \sum_{\{\pi_{1:t} | \mathcal{B}(\pi_{1:t}) = \mathcal{B}(l'_{1:s}), \pi_t = l'_s\}} \prod_{t'=1}^{t} y_{\pi_{t'}}^{t'} \log \prod_{t'=1}^{t} y_{\pi_{t'}}^{t'}
\end{aligned}
\tag{2}
$$

As in CTC [1], we allow all transitions between blank and non-blank labels and between two distinct non-blank labels. This give us the following initialization

$$\gamma(1,1) = y_b^1 \log y_b^1, \ \gamma(1,2) = y_{l_1}^1 \log y_{l_1}^1, \ \gamma(1,s) = 0, \forall s > 2. \tag{3}$$

and recursion

$$\gamma(t,s) = \bar{\gamma}(t,s)y_{l_s'}^t + \alpha(t,s) \log y_{l_s'}^t$$
$$\bar{\gamma}(t,s) = \begin{cases} \gamma(t-1,s) + \gamma(t-1,s-1) & \text{if } l_s' = b \text{ or } l_{s-2}' = l_s' \\ \gamma(t-1,s) + \gamma(t-1,s-1) + \gamma(t-1,s-2) & \text{otherwise} \end{cases} \tag{4}$$

where $\alpha$ is the forward variable defined in CTC's forward-backward algorithm.

$$\alpha(t,s) \triangleq \sum_{\{\pi_{1:t}|\mathcal{B}(\pi_{1:t})=\mathcal{B}(l_{1:s}), \pi_t=l_s\}} p(\pi_{1:t}|X)$$
$$= \sum_{\{\pi_{1:t}|\mathcal{B}(\pi_{1:t})=\mathcal{B}(l_{1:s}), \pi_t=l_s\}} \prod_{t'=1}^{t} y_{\pi_t'}^{t'} \tag{5}$$

Since all feasible paths can be divided into two groups, *i.e.* ending with the blank or ending with the last label in $l$, we get

$$Q(l) = \gamma(T, |l'|) + \gamma(T, |l'| - 1). \tag{6}$$

Similarly define the backward variable $\delta(t,s)$ as the total $p \log p$ satisfying $l'$ suffix of length $s$ at time t.

$$\delta(t,s) \triangleq \sum_{\{\pi_{t:T}|\mathcal{B}(\pi_{t:T})=\mathcal{B}(l_{s:|l'|}'), \pi_t=l_s'\}} p(\pi_{t:T}|X) \log p(\pi_{t:T}|X)$$
$$= \sum_{\{\pi_{t:T}|\mathcal{B}(\pi_{t:T})=\mathcal{B}(l_{s:|l'|}'), \pi_t=l_s'\}} \prod_{t'=t}^{T} y_{\pi_t'}^{t'} \log \prod_{t'=t}^{T} y_{\pi_t'}^{t'} \tag{7}$$

with initialization

$$\delta(T, |l'|) = y_b^T \log y_b^T, \ \delta(T, |l'| - 1) = y_{l_{|l|}}^T \log y_{l_{|l|}}^T, \ \delta(T,s) = 0, \forall s < |l'| - 1 \tag{8}$$

and recursion

$$\delta(t,s) = \bar{\delta}(t,s)y_{l_s'}^t + \beta(t,s) \log y_{l_s'}^t$$
$$\bar{\delta}(t,s) = \begin{cases} \delta(t+1,s) + \delta(t+1,s+1) & \text{if } l_s' = b \text{ or } l_{s+2}' = l_s' \\ \delta(t+1,s) + \delta(t+1,s+1) + \delta(t+1,s+2) & \text{otherwise} \end{cases} \tag{9}$$

All feasible paths can be divided into two groups, starting with the blank or starting with the first label in $l$.

$$Q(l) = \delta(0,0) + \delta(0,1). \tag{10}$$

## 1.2 Gradient Calculation

The gradient of the entropy term in EnCTC can be represented by gradient of $Q(l)$ and $p(l|X)$.

$$-\frac{\partial H(p(\pi|l,X))}{\partial y_k^t} = \frac{\partial}{\partial y_k^t}\left(\frac{Q(l)}{p(l|X)} - \log p(l|X)\right)$$
$$= \frac{1}{p(l|X)}\frac{\partial Q(l)}{\partial y_k^t} - \frac{1}{p(l|X)}\frac{\partial p(l|X)}{\partial y_k^t}\left(1 + \frac{Q(l)}{p(l|X)}\right). \tag{11}$$

Note that $\frac{\partial p(l|X)}{\partial y_k^t}$ can be calculated by CTC back-propagation algorithm, we only need to provide an algorithm for calculating $\frac{\partial Q(l)}{\partial y_k^t}$.

Since all feasible paths can be disassembled into paths going through different labels $s$ at time $t$

$$Q(l) = \sum_{s=1}^{|l'|} \sum_{\pi \in \phi_s} p(\pi|X) \log p(\pi|X)$$

$$= \sum_{s=1}^{|l'|} \sum_{\pi_{1_{1:t}} \in \phi_{1s}} \sum_{\pi_{2_{t:T}} \in \phi_{2s}} p(\pi_{1_{1:t-1}}|X) y_{l'_s}^t p(\pi_{2_{t+1,T}}|X) \log p(\pi_{1_{1:t-1}}|X) y_{l'_s}^t p(\pi_{2_{t+1,T}}|X),$$

(12)

in which $\phi$, $\phi_1$ and $\phi_2$ means

$$\phi = \{\pi | \mathcal{B}(\pi_{1:t}) = \mathcal{B}(l'_{1:s}), \mathcal{B}(\pi_{t:T}) = \mathcal{B}(l'_{s:|l'|}), \pi_t = l'_s\},$$
$$\phi_1 = \{\pi_{1:t} | \mathcal{B}(\pi_{1:t}) = \mathcal{B}(l'_{1:s}), \pi_t = l'_s\},$$
$$\phi_2 = \{\pi_{t:T} | \mathcal{B}(\pi_{t:T}) = \mathcal{B}(l'_{s:|l'|}), \pi_t = l'_s\}.$$

(13)

Consider the definition of forward-backward variables of CTC and EnCTC,

$$Q(l) = \sum_{s=1}^{|l'|} y_{l'_s}^t (\bar{\gamma}(t,s)\bar{\beta}(t,s) + \bar{\delta}(t,s)\bar{\alpha}(t,s)) + y_{l'_s}^t \log y_{l'_s}^t \bar{\alpha}(t,s)\bar{\beta}(t,s),$$

(14)

where $\bar{\alpha}(t,s)$ and $\bar{\beta}(t,s)$ are defined similarly with $\bar{\gamma}(t,s)$ and $\bar{\delta}(t,s)$.

Since $\bar{\alpha}(t,s)$, $\bar{\beta}(t,s)$, $\bar{\gamma}(t,s)$ and $\bar{\delta}(t,s)$ are constant to $y_{l'_s}^t$, the partial gradient of $Q(l)$ can be computed as

$$\frac{\partial Q(l)}{\partial y_k^t} = \sum_{s \in \text{lab}(l,k)} \bar{\gamma}(t,s)\bar{\beta}(t,s) + \bar{\delta}(t,s)\bar{\alpha}(t,s) + (1 + \log y_k^t)\bar{\alpha}(t,s)\bar{\beta}(t,s)$$

$$= \frac{1}{y_{l'_s}^{t\,2}} \sum_{s \in \text{lab}(l,k)} \gamma(t,s)\beta(t,s) + \delta(t,s)\alpha(t,s) + (1 - \log y_k^t)\alpha(t,s)\beta(t,s),$$

(15)

where $\text{lab}(l,k) = \{s : l'_s = k\}$, means the occurrence of label $k$ in the target sequence.

The partial gradient of $-H(p(\pi|l,X))$ can be computed as

$$\frac{\partial -H(p(\pi|l,X))}{\partial y_k^t} = -\frac{Q(l)}{p(l|X)^2 y_k^{t\,2}} \sum_{s \in \text{lab}(l,k)} \alpha(t,s)\beta(t,s)$$

$$+ \frac{1}{p(l|X)y_k^{t\,2}} \sum_{s \in \text{lab}(l,k)} \gamma(t,s)\beta(t,s) + \delta(t,s)\alpha(t,s) - \log y_k^t \alpha(t,s)\beta(t,s)$$

$$= -\frac{Q(l)}{p(l|X)^2 y_k^t} \sum_{\{\pi|\pi \in \mathcal{B}^{-1}(l), \pi_t = k\}} p(\pi|X)$$

$$+ \frac{1}{p(l|X)y_k^t} \sum_{\{\pi|\pi \in \mathcal{B}^{-1}(l), \pi_t = k\}} p(\pi|X) \log p(\pi|X)$$

$$= \frac{Q(l)}{p(l|X)y_k^t} \left( \frac{\sum_{\{\pi|\pi \in \mathcal{B}^{-1}(l), \pi_t = k\}} p(\pi|X) \log p(\pi|X)}{Q(l)} - \frac{\sum_{\{\pi|\pi \in \mathcal{B}^{-1}(l), \pi_t = k\}} p(\pi|X)}{p(l|X)} \right).$$

(16)

## 2 EsCTC

### 2.1 The EsCTC Forward Algorithm

We provide forward algorithm for calculating the conditional probability of EsCTC and calculating gradient with Pytorch automatic differentiation package.

$$p_\tau(l|X_{1:T}) = \sum_{z \in C_{\tau,T}(l)} \sum_{\pi \in \mathcal{B}_z^{-1}(l)} p(\pi|X_{1:T}).$$

(17)

We first define forward variable $\sigma(t_1, t_2, s)$ as the summation of the probabilities of segments $\pi_{t_1:t_2}$ that can be mapped to label $l_s$ by first removing repeated labels then removing the prefix blank(if any).

$$\sigma(t_1, t_2, s) \triangleq \sum_{\{\pi_{t_1:t_2}|\mathcal{B}(\pi_{t_1:t_2})=l_s, \pi_{t_2}=l_s\}} \prod_{t'=t_1}^{t_2} y_{\pi_{t'}}^{t'}. \tag{18}$$

In particular, $\sigma(t_1, t_2, 0)$ means the probability of an all-blank segment from $t_1$ to $t_2$. We only allow transitions from blank to label $l_s$. This give us the following initialization

$$\sigma(t, t, s) = y_{l_s}^t, \ \sigma(t, t, 0) = y_b^t, \tag{19}$$

and recursion

$$\sigma(t_1, t_2, s) = (\sigma(t_1, t_2 - 1, 0) + \sigma(t_1, t_2 - 1, s))y_{l_s}^{t_2},$$
$$\sigma(t_1, t_2, 0) = \sigma(t_1, t_2 - 1, 0)y_b^{t_2}. \tag{20}$$

Then define forward variable $\alpha_\tau(t, s)$ as the sum of probabilities of paths till time $t$ satisfying length $s$ prefix of segmentation sequences with the equal spacing coefficient $\tau$.

$$\alpha_\tau(t, s) = \sum_{z \in C_{\tau,t}(l_{1:s})} \sum_{\pi_{1:t} \in \mathcal{B}_z^{-1}(l_{1:s})} \prod_{s'=1}^{s} \prod_{t'=Z_s'}^{Z_{s'+1}-1} y_{\pi_{t'}}^{t'}. \tag{21}$$

When $s = 1$, $\alpha_\tau(t, s)$ degenerates to a single segment probability similar to Equation 18. This give us the initialization

$$\alpha_\tau(t, 1) = \begin{cases} \sigma(1, t, 1) & \text{if } t \leq \tau \\ 0 & \text{otherwise} \end{cases} \tag{22}$$

To limit the length of each segment and use blanks to separate the same labels, we have recursion

$$\alpha_\tau(t, s) = \begin{cases} \sum_{t'=1}^{\tau\frac{T}{|l|}} \alpha_\tau(t - t', s - 1)\sigma(t - t' + 1, t, s) & \text{if } l_{s-1} \neq l_s \\ \sum_{t'=2}^{\tau\frac{T}{|l|}} \alpha_\tau(t - t', s - 1)y_b^{t-t'+1}\sigma(t - t' + 2, t, s) & \text{otherwise} \end{cases} \tag{23}$$

The complete correspondence between input and output sequences includes segment ending with $l_{|l|}$ and a full blank segment with length not exceeding $\tau\frac{T}{|l|}$

$$p_\tau(l|X_{1:T}) = \alpha_\tau(T, |l|) + \sum_{t'=1}^{\tau\frac{T}{|l|}} \alpha_\tau(T - t', |l|)\sigma(T - t' + 1, T, 0). \tag{24}$$

# 3 EnEsCTC

## 3.1 The EnEsCTC Forward Algorithm

We provide a forward algorithm for calculating the entropy term of EnEsCTC and calculating the gradient with Pytorch automatic differentiation package.

$$H(p_\tau(\pi|l, X)) = - \sum_{z \in C_{\tau,T}(l)} \sum_{\pi \in \mathcal{B}_z^{-1}(l)} p(\pi|l, X) \log p(\pi|l, X). \tag{25}$$

The entropy term in EnCTC can be converted to

$$H(p_\tau(\pi|l, X)) = -\frac{1}{p_\tau(l|X)} \sum_{z \in C_{\tau,T}(l)} \sum_{\pi \in \mathcal{B}_z^{-1}(l)} p(\pi|X) \log p(\pi|X) + \log p_\tau(l|X). \tag{26}$$

Since $p_\tau(l|X)$ can be computed by EnCTC forward-backward algorithm, here we only need to provide dynamic programming algorithms for calculating $Q_\tau(l) = \sum_{z \in C_{\tau,T}(l)} \sum_{\pi \in \mathcal{B}_z^{-1}(l)} p(\pi|X) \log p(\pi|X)$.

Similar to EsCTC, we first define $\eta(t_1, t_2, s)$ as the sum $p \log p$ of segments $\pi_{t_1:t_2}$ that can be mapped to label $l_s$ by first removing repeated labels then removing the prefix blank (if any).

$$\eta(t_1, t_2, s) \triangleq \sum_{\{\pi_{t_1:t_2} | \mathcal{B}(\pi_{t_1:t_2}) = l_s, \pi_{t_2} = l_s\}} \prod_{t'=t_1}^{t_2} y_{\pi_{t'}}^{t'} \log \prod_{t'=t_1}^{t_2} y_{\pi_{t'}}^{t'}, \tag{27}$$

with initialization

$$\eta(t, t, s) = y_{l_s}^t \log y_{l_s}^t$$
$$\eta(t, t, 0) = y_b^t \log y_b^t \tag{28}$$

and recursion

$$\eta(t_1, t_2, s) = (\sigma(t_1, t_2 - 1, 0) + \sigma(t_1, t_2 - 1, s)) y_{l_s}^{t_2} + \sigma(t_1, t_2, s) \log y_{l_s}^{t_2}$$
$$\eta(t_1, t_2, 0) = \sigma(t_1, t_2 - 1, 0) y_b^{t_2} + \sigma(t_1, t_2, 0) \log y_b^{t_2}. \tag{29}$$

Then define forward variable $\gamma_\tau(t, s)$ as the sum $p \log p$ of paths till time $t$ satisfying length $s$ prefix of segmentation sequences with the equal spacing coefficient $\tau$.

$$\gamma_\tau(t, s) = \sum_{z \in C_{\tau, t}(l_{1:s})} \sum_{\pi_{1:t} \in \mathcal{B}_z^{-1}(l_{1:s})} p(\pi_{1:t} | X) \log p(\pi_{1:t} | X)$$

$$= \sum_{z \in C_{\tau, t}(l_{1:s})} \sum_{\pi_{1:t} \in \mathcal{B}_z^{-1}(l_{1:s})} \prod_{s'=1}^{s} \prod_{t'=Z_s'}^{Z_{s'+1}-1} y_{\pi_{t'}}^{t'} \log \prod_{s'=1}^{s} \prod_{t'=Z_s'}^{Z_{s'+1}-1} y_{\pi_{t'}}^{t'}, \tag{30}$$

with initialization

$$\gamma_\tau(t, 1) = \begin{cases} \eta(1, t, 1) & \text{if } t \leq \tau \\ 0 & \text{otherwise} \end{cases} \tag{31}$$

and recursion

$$\gamma_\tau(t, s) = \begin{cases} \sum_{t'=1}^{\tau \frac{T}{|l|}} \gamma_\tau(t - t', s - 1)\sigma(t - t' + 1, t, s) + \alpha_\tau(t - t', s - 1)\eta(t - t' + 1, t, s) \\ \qquad \text{if } l_{s-1} \neq l_s \\ \sum_{t'=2}^{\tau \frac{T}{|l|}} \gamma_\tau(t - t', s - 1)y_b^{t-t'+1}\sigma(t - t' + 2, t, s) + \\ \qquad \alpha_\tau(t - t', s - 1)y_b^{t-t'+1}\eta(t - t' + 2, t, s) + \\ \qquad \alpha_\tau(t - t', s - 1)y_b^{t-t'+1} \log y_b^{t-t'+1}\sigma(t - t' + 2, t, s) \\ \qquad \text{otherwise} \end{cases} \tag{32}$$

For the complete correspondence between input and output sequences,

$$Q_\tau(l) = \gamma_\tau(T, |l|) + \sum_{t'=1}^{\tau \frac{T}{|l|}} \gamma_\tau(T - t', |l|)\sigma(T - t' + 1, T, 0)$$
$$+ \alpha_\tau(T - t', |l|)\sigma(T - t' + 1, T, 0) \log \sigma(T - t' + 1, T, 0). \tag{33}$$

## 4 Path Pruning Analysis for EsCTC

Table 1: The influence of $\tau$ on path pruning for different data scales.

| T=26 | $|l|=4$ | $|l|=8$ | $|l|=12$ | $|l|=16$ | T=52 | $|l|=4$ | $|l|=8$ | $|l|=12$ | $|l|=16$ |
|---|---|---|---|---|---|---|---|---|---|
| CTC Path | 4e6 | 2e9 | 1e10 | 4e8 | CTC Path | 1e9 | 1e14 | 1e17 | 3e19 |
| $\tau = 1.0$ | ×0.13 | ×0.04 | ×0.07 | ×0.04 | $\tau = 1.0$ | ×0.06 | ×7e-4 | ×4e-4 | ×2e-4 |
| $\tau = 1.2$ | ×0.31 | ×0.04 | ×0.07 | ×0.04 | $\tau = 1.2$ | ×0.19 | ×0.013 | ×0.021 | ×0.03 |
| $\tau = 1.5$ | ×0.66 | ×0.56 | ×0.47 | ×0.60 | $\tau = 1.5$ | ×0.62 | ×0.17 | ×0.13 | ×0.20 |
| $\tau = 2.0$ | ×0.95 | ×0.89 | ×0.93 | ×0.92 | $\tau = 2.0$ | ×0.92 | ×0.70 | ×0.69 | ×0.68 |
| T=104 | $|l|=4$ | $|l|=8$ | $|l|=12$ | $|l|=16$ | T=208 | $|l|=4$ | $|l|=8$ | $|l|=12$ | $|l|=16$ |
| CTC Path | 3e11 | 8e18 | 4e24 | 1e29 | CTC Path | 8e13 | 6e23 | 7e31 | 5e38 |
| $\tau = 1.0$ | ×0.03 | ×3e-4 | ×0.00 | ×0.00 | $\tau = 1.0$ | ×0.02 | ×3e-5 | ×0.00 | ×0.00 |
| $\tau = 1.2$ | ×0.16 | ×0.008 | ×3e-5 | ×3e-5 | $\tau = 1.2$ | ×0.13 | ×0.005 | ×7e-5 | ×5e-6 |
| $\tau = 1.5$ | ×0.52 | ×0.21 | ×0.02 | ×0.01 | $\tau = 1.5$ | ×0.49 | ×0.12 | ×0.03 | ×0.02 |
| $\tau = 2.0$ | ×0.89 | ×0.68 | ×0.42 | ×0.38 | $\tau = 2.0$ | ×0.88 | ×0.61 | ×0.39 | ×0.32 |

# References

[1] Alex Graves and Faustino Gomez. Connectionist temporal classification:labelling unsegmented sequence data with recurrent neural networks. In *International Conference on Machine Learning (ICML)*, pages 369–376, 2006.