[Reviews · NeurIPS 2018]

Reviewer 1



This work presents a method for end-to-end sequence learning, and more specifically in the framework of Connectionist Temporal Classification (CTC). The paper has two main contributions: - The first is a regularization of the training of the CTC objective in order to reduce the over-confidence of the model. In order to do that, the authors propose a method based on conditional entropy. More specifically, the proposed regularization would encourages the model to explore paths that are close to the dominant one. - The second contribution is a pruning method in order to reduce the space of possible alignments. In order to do so, they suppose that the consecutive elements of a sequence have equal spacing. The authors prove that this situation corresponds to the maximum of entropy over the possible segmentations of the sequence. The paper mainly advocates for the combination of the two contributions. The authors also provide details about dynamic programming to optimize the proposed objective. They finally compare their methods to the state-of-the-art approaches on different datasets. The contributions of the paper are well motivated, and their effect seems to be significant according to the experimental results. The two proposed regularization are also well explained intuitively, and through supporting theory. In addition, the effect of the regularization is well analyzed in the experiments section. For all these reasons I vote for acceptance of this paper. Minor comment: There is a typo in equation (3) in the partial derivative notation.

Reviewer 2



The paper proposes to regularize the CTC classification loss by increasing the entropy of the probability assigned to alignment paths, rather than to individual outputs of the network. The paper is well motivated, provides clean intuitions and good empirical results. I only have minor comments about the paper: - The quality of Fig. 2 can be improved - i assume the plots show the posterior of different symbols, can a legend be added to make this more explicit? Possible addition to the paper's related work section: The authors consider bidirectional networks and therefore do not have the problem of the network delaying its outputs until enough future context is seen. This is often impractical in ASR systems and constraints to the path taken by CTC were considered to alleviate this issue - this is in line with the proposed equal spacing prior (see [1]). It is also possible to sample CTC paths and optimize the network outputs for them, rather than doing the full forward backward algorithm [2]. Path sampling has a side-effect of also improving the accuracy of the model, possibly by introducing a smoothing behavior. I believe referring to the two above works will strengthen the motivation behind the proposed entropy regularization. [1] https://static.googleusercontent.com/media/research.google.com/en//pubs/archive/44269.pdf [2] https://storage.googleapis.com/pub-tools-public-publication-data/pdf/b0ffb206352c9f77a03d6a1df224a8240bce974a.pdf

Reviewer 3



This paper presents a entropy regularization technique for the CTC algorithm. The basic idea is to make sure the entropy for the space under the collapsing constraint is not too small. This prevents the tendency that CTC is likely to choose one path and give small probabilities to alternatives. Entropy regularization is pretty common technique. Though not very exciting, this paper applies the entropy regularization to CTC loss and it seems to be a nice addition to the CTC related technique. I am slightly confused with the theorem 3.1. according to your theorem, do you really need to enforce equal space constraints directly. does the regularization itself sort of do that already?